# Favipiravir Inhibits Hepatitis A Virus Infection in Human Hepatocytes

**DOI:** 10.3390/ijms23052631

**Published:** 2022-02-27

**Authors:** Reina Sasaki-Tanaka, Toshikatsu Shibata, Hiroaki Okamoto, Mitsuhiko Moriyama, Tatsuo Kanda

**Affiliations:** 1Division of Gastroenterology and Hepatology, Department of Medicine, Nihon University School of Medicine, 30-1 Oyaguchi-kamicho, Itabashi-ku, Tokyo 173-8610, Japan; toshimei@gmail.com (T.S.); moriyama.mitsuhiko@nihon-u.ac.jp (M.M.); kanda.tatsuo@nihon-u.ac.jp (T.K.); 2Division of Virology, Department of Infection and Immunity, Jichi Medical University School of Medicine, 3311-1 Yakushiji, Shimotsuke-shi 329-0498, Japan; hokamoto@jichi.ac.jp

**Keywords:** hepatitis A virus, favipiravir, ribavirin, internal ribosomal entry site, mutagenesis

## Abstract

Hepatitis A virus (HAV) is a causative agent of acute hepatitis and can occasionally induce acute liver failure. However, specific potent anti-HAV drug is not available on the market currently. Thus, we investigated several novel therapeutic drugs through a drug repositioning approach, targeting ribonucleic acid (RNA)-dependent RNA polymerase and RNA-dependent deoxyribonucleic acid polymerase. In the present study, we examined the anti-HAV activity of 18 drugs by measuring the HAV subgenomic replicon and HAV HA11-1299 genotype IIIA replication in human hepatoma cell lines, using a reporter assay and real-time reverse transcription polymerase chain reaction, respectively. Mutagenesis of the HAV 5’ untranslated region was also examined by next-generation sequencing. These specific parameters were explored because lethal mutagenesis has emerged as a novel potential therapeutic approach to treat RNA virus infections. Favipiravir inhibited HAV replication in both Huh7 and PLC/PRF/5 cells, although ribavirin inhibited HAV replication in only Huh7 cells. Next-generation sequencing demonstrated that favipiravir could introduce nucleotide mutations into the HAV genome more than ribavirin. In conclusion, favipiravir could introduce nucleotide mutations into the HAV genome and work as an antiviral against HAV infection. Provided that further in vivo experiments confirm its efficacy, favipiravir would be useful for the treatment of severe HAV infection.

## 1. Introduction

The hepatitis A virus (HAV) genome consists of a single-stranded, positive-sense RNA, approximately 7.5 kilobases in length, encoding a large open reading frame (ORF) that is flanked by highly conserved 5′ and 3′ untranslated regions (UTRs), and encodes four structural proteins (VP4, VP2, VP3, and VP1) and seven nonstructural proteins (2A, 2B, 2C, 3A, 3B, 3C and 3D). Initiation translation of HAV RNA is mediated through the HAV internal ribosome entry site (IRES) element located mainly in the 5’ UTR. It is reported that the HAV 3D protein is a viral ribonucleic acid (RNA)-dependent RNA polymerase (RdRp), and HAV replicates through genomic plus-stranded RNA [1].

In general, individuals with HAV infection recover with or without intervention. However, in cases of HAV-associated acute liver failure (ALF) (0.015–0.5%), intensive care, including urgent liver transplantation, is required [2]. Reported risk factors for severe acute hepatitis or for higher mortality induced by HAV infection include an age of more than 40 years, preexisting liver disease, diabetes mellitus, and cardiovascular disease [3,4]. Adjusted odds ratios for death by age 40–59 years and age over 60 years are 7.89 and 14.88, respectively, compared to age 0–19 years [3]. Adjusted odds ratios for death by preexisting nonviral liver disease, history of hepatitis B, diabetes, and cardiovascular disease are 5.2, 2.4, 2.2, and 2.2, respectively [4].

As of 2020, it is estimated that there are 728 million persons aged 65 years or older worldwide, and this number is expected to increase to 1.5 billion older persons by 2050 [5]. Thus, it is possible that the number of patients with HAV-associated ALF will increase in the near future.

As of 2021, over 460 million people have diabetes mellitus worldwide, and in some regions, this number is predicted to more than double by 2045 [6]. Additionally, the number of persons without HAV immunity will increase with improvements in hygiene [7]; thus, there is no doubt that antiviral therapies for HAV infection will be urgently required. As these factors also may make it possible that the number of patients with severe HAV infection could increase, it is important to take appropriate measures promptly against HAV-associated ALF, including specific antiviral therapies for HAV infection.

Although there have been several reports about specific antiviral therapies for HAV infection, both direct-acting antiviral agents (DAAs) and host-targeting agents (HTAs) to control effectively HAV infection should continue to be explored [8,9]. Small interfering RNAs against HAV and HAV 3C cysteine protease inhibitors are promising DAAs against HAV [8,10]. Interferons, ribavirin, and amantadine are also reported as broad-target HTAs against HAV infection [8,11,12]. It is unknown whether these drugs have enough effects in clinical settings, and specific and potent anti-HAV drug is not available on the market. The development of drugs for HAV infection is challenging now since there are an estimated 170 million new cases of acute hepatitis A [13] and HAV is still the most common cause of acute viral hepatitis [14].

In the present study, therefore, we investigated potentially effective drugs by drug repositioning. We examined the anti-HAV activity of these 18 drugs, including 6 RdRp inhibitors and 12 RNA-dependent deoxyribonucleic acid (DNA) polymerase (RdDp) inhibitors, by measuring the HAV subgenomic replicon and HAV replication. To explore the mechanism of action of the selected drugs, we also examined the mutagenesis of the HAV 5’ UTR, using next-generation sequencing methods. Updated drugs for acute hepatitis A are needed since improved cure rates of acute HAV infection are critical to create strategies for global intervention.

## 2. Results

### 2.1. Favipiravir, Ribavirin, and Foscarnet Sodium Significantly Downregulate Hepatitis A Virus Subgenomic Replicon Replication in HuhT7 Cells

We first evaluated the cytotoxic effects of 18 polymerase inhibitors on HuhT7 cells by dimethylthiazol carboxymethoxyphenyl sulfophenyl tetrazolium (MTS) assays. Cell viabilities following treatment with the 18 drugs were examined and compared to dimethyl sulfoxide (DMSO)-treated controls. As depicted in Figure 1A–R, the 18 polymerase inhibitors did not impact HuhT7 cell viability at concentrations of 10 μM or lower.

Favipiravir, ribavirin, and foscarnet sodium treatment significantly downregulated HAV subgenomic replicon replication in a dose-dependent manner (Figure 2A–R). Favipiravir treatment at a concentration of 1 μM resulted in a >60% reduction in HAV subgenomic replicon replication in HuhT7 cells, whereas 10 μM favipiravir was required for a 67% decrease in HAV subgenomic replicon replication (Figure 2A). Treatment with ribavirin and foscarnet sodium at a concentration of 1 μM resulted in a 53% and a 53% reduction, respectively, in HAV subgenomic replicon replication, whereas 10 μM ribavirin and foscarnet sodium was required for a 66% and a 58% decrease, respectively, in HAV subgenomic replicon replication (Figure 2B,G). Valacyclovir hydrochloride treatment at a concentration of 10 μM resulted in a 64% reduction in HAV subgenomic replicon replication (Figure 2H). No significant inhibition was observed in HAV-transfected Huh7 cells treated with the other drugs (Figure 2C–F,I–R). As we used valacyclovir hydrochloride and adefovir dipivoxil prodrugs instead of the directly active compounds acyclovir and adefovir, it is possible that these ester prodrugs could not be properly hydrolyzed in HuhT7 cells. Our results are in agreement with earlier observations suggesting that 100 μM ribavirin inhibits HAV replication [15].

### 2.2. Favipiravir and Ribavirin Significantly Downregulate Hepatitis A Virus HA11-1299 Genotype IIIA Replication in Huh7 Cells

As depicted in Figure 3A–C, favipiravir, ribavirin, and foscarnet sodium did not impact Huh7 cell viability at concentrations of 100 μM or lower. As shown in Figure 3D, HAV RNA was significantly reduced upon treatment with favipiravir in HAV-infected Huh7 cells. Favipiravir treatment at a concentration of 10 μM resulted in a 29% reduction, whereas 100 μM resulted in an 80% reduction in HAV RNA levels. Favipiravir inhibited HAV replication in a dose-dependent manner, with an estimated half maximal inhibitory concentration (IC_50_) of 18.8 μM. Ribavirin treatment at a concentration of 100 μM resulted in a >25% reduction in the expression of HAV replication in Huh7 cells (Figure 3E), whereas 100 μM foscarnet sodium showed no evidence of inhibition of HAV replication in HAV-infected Huh7 cells (Figure 3F).

We also examined whether HAV RNA levels could be inhibited by favipiravir or ribavirin in PLC/PRF/5 cells. Favipiravir at a concentration of 100 μM resulted in a 30–40% reduction in the HAV RNA levels in PLC/PRF/5 cells, whereas HAV RNA levels were not reduced by 100 μM ribavirin in PLC/PRF/5 cells.

### 2.3. Favipiravir Increases the Mutation Frequency of the Hepatitis A Virus Genome Sequence

The amplified length was 469 bp, and the target site was located within the 5’ UTR of the HAV genome. The total numbers of reads of untreated control cells and those of cells treated with favipiravir, ribavirin, and foscarnet sodium were 85,107, 78,235, 86,047, and 82,399, respectively. The percentages of mutation reads identified in the number of total reads of untreated control cells and those in the sequences of cells treated with favipiravir, ribavirin, and foscarnet sodium were 22.8, 32.8, 22.1, and 24.3%, respectively (Table 1).

This analysis revealed that the percentage of total mutations in isolates from HAV-infected Huh7 cells treated with favipiravir was larger than the percentage of total mutations in other isolates. There was no significant change in the percentage of total mutations between control and HAV-infected Huh7 cells treated with ribavirin and foscarnet sodium. The percentages of consensus sequences (Seq 1) of the control cells and those treated with favipiravir, ribavirin, or foscarnet sodium were 77.2, 67.2, 77.9, and 75.7%, respectively (Table 1). The consensus sequences of isolates from HAV-infected Huh7 cells and the distribution of mutations found in the HAV 5’ UTR with 100 μM favipiravir, 100 μM ribavirin, and 100 μM foscarnet sodium are shown in Figure 4. There were no specific mutational hotspots, and almost all mutations were single-nucleotide mutations. Our results demonstrated that there were no specific single-nucleotide variations in the HAV 5’ UTR during favipiravir, ribavirin, and foscarnet sodium treatment.

## 3. Discussion

In the present study, we focused on the effects of 6 RdRp inhibitors and 12 RdDp inhibitors on HAV replication to investigate therapeutic antiviral drugs through drug repositioning. We demonstrated the anti-HAV activity of favipiravir and ribavirin by measuring the HAV subgenomic replicon replication in HuhT7 and HAV replication in Huh7 cells. We also investigated the mutagenesis of the HAV 5’ UTR using next-generation sequencing methods. We found that favipiravir decreased HAV replication in a dose-dependent manner and induced nucleotide mutations in the HAV genome (Figure 3, Table 1).

As a catalytic domain of RdRp is widely conserved among RNA viruses [16], the RdRp of viruses is considered to play a pivotal role in the replication cycle of most RNA viruses. Thus, it is a promising therapeutic target for developing antiviral agents against RNA viruses. Moreover, some RdDps have phylogenetic and structural similarities to viral RdRps [17]. Previous reports suggest that antiviral mutagenesis could be an effective approach to eliminate pathogenetic viruses [18,19,20].

Previous studies demonstrated that ribavirin was a promising candidate for acute hepatitis A [15,21], although the mechanism of its action for HAV infection is not well understood. Ribavirin is an artificial guanosine nucleoside analog (Figure 5A) that is used to treat various viral infection, including Lassa fever, respiratory syncytial virus, and HCV infections [22,23,24,25]. The IC_50_ values of ribavirin for respiratory syncytial virus, HCV, and parainfluenza-3 virus were 20.9 μM, 12.8 μM, and 197.9 μM, respectively [19,26]. In the present study, the use of 100 μM ribavirin did not show an IC_50_ (Figure 3). The serum concentration of ribavirin in complete responders during the combination of interferon alfa-2b and ribavirin therapy for chronic hepatitis C was reported to be 2000–3000 ng/mL (8–12 μM) [27]. We may need to use a much higher concentration of ribavirin for HAV infection in vitro. As a higher serum concentration of ribavirin could lead to adverse events, such as hemolytic anemia, it may be difficult for humans to use a higher dose of ribavirin. Ribavirin inhibits host inosine monophosphate dehydrogenase (IMPDH), which lowers intracellular GTP pools [28]. Moreover, ribavirin is a mutagenic nucleoside analog in rapidly replicating viruses, inducing error catastrophes and competing with physiological nucleotides for the HCV RdRp active site [29]. Ribavirin also induces lethal mutagenesis and error-prone HCV replication [30,31,32]. We have reported previously that A→G and G→A mutations increased in comparison with the total number of transition mutations [32]. Ribavirin also induces lethal mutagenesis of poliovirus [33]. In the present study, there was no upregulation of the percentage of mutations or specific single-nucleotide variants during ribavirin treatment in HAV infection (Table 1). The effects of 100 μM ribavirin on HAV RNA levels were not observed in PLC/PRF/5 cells. More incubation time might be required.

Favipiravir has been shown to have a broad spectrum of effects against RNA viruses, such as Influenza virus, Ebola virus (EBOV), Crimean–Congo hemorrhagic fever virus, Lassa virus, Rift Valley fever virus, hemorrhagic fever arenavirus, Chikungunya virus, and Norovirus [34]. Moreover, favipiravir was effective in reducing viral replication in a recent outbreak of coronavirus disease 2019 (COVID-19) caused by the novel coronavirus designated severe acute respiratory syndrome coronavirus 2 (SARS-CoV-2), and the half-maximal effective concentration (EC_50_) was 61.88 μM in vitro [35]. The IC_50_ values for favipiravir against Zika virus and Crimean–Congo hemorrhagic fever were 35 ±14 μM and 1.1 µg/mL, respectively [36,37]. Oral favipiravir (300 mg/kg dosed once a day) led to 43.5% survival of Ebola virus infection in guinea pigs [38]. The median serum concentration of favipiravir among COVID-19 patients treated with favipiravir was 35.22–60.85 μg/mL (224–387 μM) [39]. In the present study, we found that the IC_50_ of favipiravir was 18.8 μM for HAV infection (Figure 3). It might be useful and safe for humans to use the present dose of favipiravir. Favipiravir is also a mutagenic nucleoside analog (Figure 5B), and viral RNA polymerase mistakenly recognizes favipiravir–RTP as a purine nucleotide [40]. Increases in C → U or A → G minority single-nucleotide variants during favipiravir treatment have been reported [41,42]. Another group reported that favipiravir leads to an excess of G → A and C → U transitions as lethal mutagenesis into HCV populations and works as an antiviral against HCV [43]. In the present study, the percentage of mutation was elevated by 10% compared to the control (Table 1); however, there were no specific single-nucleotide variants in the HAV 5’ UTR during ribavirin or favipiravir treatment (Figure 4). It is possible that favipiravir may induce lethal mutations in the HAV genome and decrease HAV replication.

There may be some differences in the mechanism of HAV inhibition between favipiravir and ribavirin. Arias et al. showed that favipiravir and ribavirin caused significant increases in the mutation frequencies of replicating murine norovirus [44]. However, there was some difference in that favipiravir inhibition of norovirus occurred in a gradual manner, but ribavirin inhibition occurred from an early time point (8 h). Moreover, favipiravir induced many more mutations (five- to six-fold) than ribavirin (three-fold). Another difference was in viral resistance to the drugs; ribavirin monotherapy tends to lead to NS5B F415Y mutation in HCV genotype 1a RNA, and it represents a ribavirin-resistant variant [45], and a single mutation (G64S) in RdRp of poliovirus confers resistance to ribavirin [46], while favipiravir is unlikely to lead to resistance during the treatment of influenza viruses because favipiravir works as a chain terminator [34,47]. Favipiravir works as a chain terminator of poliovirus also, but resistant mutants that were isolated in ribavirin treatment were not detected [34]. The present study supports these previous studies and provides similar observations of HAV infection. In the present study, favipiravir or ribavirin treatment resulted in a significant reduction in HAV RNA levels (Figure 3). Moreover, our data also indicate that favipiravir’s inhibitory effect on HAV replication is stronger than that of ribavirin. Some differences are observed between favipiravir and ribavirin in the percentage of mutation (32.8 and 22.1%, respectively) and the total number of HAV 5’ UTR sequences also decreased with favipiravir treatment of HAV-infected hepatocytes (Table 1), suggesting that favipiravir and ribavirin may have different mechanisms of action. Other groups have suggested the synergistic lethal mutagenesis of favipiravir and ribavirin in response to HCV infection [48]. On this point, further investigation is needed.

In the present study, Huh7 and its derived cells were used for most of the experiments. As this may be one of the study’s limitations, another cell line would be useful for further study [49]. As another limitation of the present study, we did not examine the direct effects of favipiravir on HAV infection using a plaque assay [50,51]. Further studies focusing on the mutagenesis of HAV polymerase genes and the meaning of these mutations, and in vivo evaluation of favipiravir administration in animal models, will also be important to assess the therapeutic effects of favipiravir. After this evidence is established, we may examine whether favipiravir is effective for HAV infection with safety.

## 4. Materials and Methods

### 4.1. Cell Lines and Reagents

The human hepatoma cell lines Huh7, PLC/PRF/5, and HuhT7, a stably transformed derivative of Huh7 expressing T7 RNA polymerase, were used. Huh7 and HuhT7 cells were kindly provided by Prof. Bartenschlager and Prof. Gauss-Müller, respectively [52,53]. PLC/PRF/5 cells were purchased from the National Institutes of Biomedical Innovation, Health and Nutrition JCRB Cell Bank (Ibaraki, Osaka, Japan) [49]. Cells were maintained in Roswell Park Memorial Institute medium (RPMI; Sigma-Aldrich, St. Louis, MO, USA) containing 10% heat-inactivated fetal bovine serum (FBS; Sigma-Aldrich), 100 units/mL penicillin, and 100 μg/mL streptomycin (Sigma-Aldrich) under 5% CO_2_ at 37 °C. The HAV HA11-1299 genotype IIIA was used for HAV infection in the present study [54]. The replication-competent HAV subgenomic replicon pT7-18f-LUC contains an ORF of firefly luciferase flanked by the first four amino acids of the HAV polyprotein and by 12 C-terminal amino acids of VP1. These segments are followed by the P2 and P3 domains of the HAV polyprotein (HAV strain HM175 18f) [53].

We selected 6 RdRp inhibitors and 12 RdDp inhibitors. The 6 RdRp inhibitors were (i) favipiravir, which is applied to treat influenza virus infection [55]; (ii) ribavirin, a synthetic nucleoside analog of ribofuranose with activity against hepatitis C virus (HCV) and other RNA viruses [56]; (iii) clemizole, an H1 histamine receptor antagonist that inhibits HCV NS4B RNA binding and HCV replication [57]; (iv) lomibuvir, an inhibitor of HCV polymerase [58]; (v) PSI-6206, a sofosbuvir metabolite and selective HCV RNA polymerase inhibitor [59]; and (vi) sofosbuvir, a uridine monophosphate analog inhibitor of HCV polymerase NS5B [60].

The 12 RdDp inhibitors were (i) foscarnet sodium, an antiviral agent used in the treatment of cytomegalovirus retinitis and human herpesviruses and human immunodeficiency virus (HIV) infections [61,62]; (ii) valacyclovir hydrochloride, an acyclovir prodrug that inhibits viral DNA replication after metabolization [63]; (iii) vidarabine, a nucleoside antibiotic isolated from Streptomyces antibiotics that has some antineoplastic properties and has broad-spectrum activity against DNA viruses [64]; (iv) oltipraz, a synthetic dithiolethione with potential chemopreventive and anti-angiogenic properties and an inhibitor of HIV-1 replication by inactivating reverse transcriptase [65]; (v) zalcitabine, a nucleoside analog reverse transcriptase inhibitor (NRTI) and inhibitor of HIV replication by binding to reverse transcriptase terminated synthesis of viral DNA [66]; (vi) clevudine, a synthetic pyrimidine analog with activity against hepatitis B virus (HBV) replication [67]; (vii) famciclovir, a herpes simplex virus (HSV) nucleoside analog DNA polymerase inhibitor [68]; (viii) tenofovir, an adenine analog reverse transcriptase inhibitor with antiviral activity against HIV-1 and HBV [69]; (ix) salicylanilide, a group of compounds with antiviral potency and antibacterial and antifungal activities [70]; (x) adefovir dipivoxil, a dipivoxil formulation of adefovir and a nucleoside reverse transcriptase inhibitor analog of adenosine with activity against HBV, herpes virus, and HIV [71]; (xi) entecavir hydrate, a entecavir deoxyguanine nucleoside analog and inhibitor of HBV replication [72]; and (xii) lamivudine, a reverse transcriptase inhibitor in which a sulfur atom replaces the 3’ carbon of the pentose ring for HBV and HIV infection [73]. These 18 drugs were purchased from TargetMol (Wellesley Hills, MA, USA).

### 4.2. Transfection of the HAV Subgenomic Replicon into HuhT7 Cells and Reporter Assays

Twenty-four hours prior to transfection, HuhT7 cells (approximately 1 × 10^5^ cells/well) were placed in a 24-well plate (Iwaki Glass, Tokyo, Japan). Cells were transiently transfected with 0.2 μg HAV subgenomic replicon using Effectene Transfection Reagent (Qiagen, Hilden, Germany) in accordance with the manufacturer’s protocol. Twenty-four hours after transfection, cells were treated with 0, 1, and 10 μM of the 18 drugs. After 72 h of transfection, cells were harvested using reporter lysis buffer (Toyo Ink, Tokyo, Japan), and firefly luciferase activity was determined using a Luminescencer JNR II AB-2300 (ATTO, Tokyo, Japan). Firefly luciferase activities were compared to DMSO-treated controls.

### 4.3. Infection of Huh7 Cells with HAV

Twenty-four hours prior to infection, Huh7 cells (approximately 3 × 10^5^ cells/well) were placed in 6-well plates (Iwaki Glass). Cells were washed twice with phosphate-buffered saline (PBS) and infected with the HAV HA11-1299 genotype IIIA strain at a multiplicity of infection (MOI) of 0.1 in serum-free RPMI. The HAV inoculum was incubated with hepatocytes for 6 h, and 1 mL of RPMI containing 2% FBS was added. After 24 h of infection, cells were washed once with PBS, followed by the addition of 1 mL of RPMI containing 5% FBS. Then, cells were treated with favipiravir, ribavirin, or foscarnet sodium, which suppressed HAV replication in the HAV subgenomic replicon assay, at 0, 1, 10, and 100 μM for 72 h. After 96 h of infection, HAV RNA levels were determined by real-time reverse transcription polymerase chain reaction (RT-PCR) as described below.

### 4.4. RNA Extraction and Quantification of HAV RNA

Total cellular RNA was extracted using the RNeasy Mini Kit (Qiagen) according to the manufacturer’s instructions. Complementary DNA (cDNA) was synthesized with oligo dT primers and random hexamers using the PrimeScript RT reagent kit (Perfect Real Time; Takara, Otsu, Shiga, Japan). Reverse transcription was performed at 37 °C for 15 min, followed by 95 °C for 5 s. Quantitative amplification of cDNA was monitored with SYBR Green by real-time RT-PCR in a QuantStudio 3 real-time RT-PCR System (Thermo Fisher Scientific, Waltham, MA, USA). Thermal cycling conditions were 95 °C for 10 min followed by 40 cycles at 95 °C, 15 s for denaturation, and 1 min at 60 °C for annealing and extension. Data analysis was based on the ddCt method. Specificity was validated using melting curve analysis. The primer sets are shown in Table 2.

### 4.5. Dimethylthiazol Carboxymethoxyphenyl Sulfophenyl Tetrazolium (MTS) Assays

For the evaluation of cell viability, MTS assays were performed using the CellTiter 96 AQueous One Solution Cell Proliferation Assay (Promega, Madison, WI, USA). Enzyme activity was measured with a Bio-Rad iMark microplate reader (Bio-Rad, Hercules, CA, USA) at 490 nm. Cell viabilities were compared to DMSO-treated controls.

### 4.6. Targeted Deep Sequencing

It was previously reported that the antiviral effect of favipiravir and ribavirin was correlated with the incorporation of a large number of mutations into viral genomes in other viruses [36,74,75]. To explain the mechanisms of HAV inhibition by favipiravir and ribavirin, we examined nucleotide mutations in the HAV genome by next-generation sequencing. We first extracted cellular RNA from HAV-infected Huh7 cells treated with or without favipiravir, ribavirin, or foscarnet sodium at 100 μM each for 72 h. We next amplified the target site by linker-added specific primers (Table 2) using a PrimeScript II High Fidelity One Step RT-PCR Kit (Takara). Each reaction was performed at 45 °C for 10 min and 94 °C for 2 min, followed by 45 cycles at 98 °C, 10 s for denaturation, 1 min at 60 °C for annealing, and 10 s at 68 °C for extension. The products were purified using a QIAquick PCR purification kit (Qiagen). Targeted deep sequencing was performed using an Illumina MiSeq System (Illumina K.K., Tokyo, Japan) at the instruction of Hokkaido System Science Co. Ltd. (Sapporo, Hokkaido, Japan).

### 4.7. Calculation of the Half Maximal Inhibitory Concentration (IC_50_)

The concentrations of each analog that produce 50% of a maximal inhibition of HAV are IC_50_, which are obtained from the following equation: IC_50_ = 10^[LOG(A/B) × (50 − C)/(D − C) + LOG(B)]. Variables indicate a higher concentration of two values that sandwich IC_50_ (A), a lower concentration of two values that sandwich IC_50_ (B), HAV RNA levels (%) at B (C), and HAV RNA levels (%) at A (D) (Figure 6).

### 4.8. Statistical Analysis

Data are expressed as the means ± standard deviations (SD). Statistical analyses were performed with Student’s *t* test. *p* < 0.05 was considered significant. All assays were performed in triplicate.

## 5. Conclusions

We demonstrated that favipiravir effectively suppressed HAV replication through the introduction of mutagenesis into the HAV genome and that favipiravir could be useful for the control of HAV infection.

## Figures and Tables

**Figure 1 ijms-23-02631-f001:**
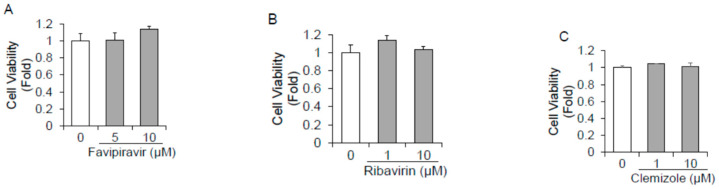
Cytotoxicity of 18 polymerase inhibitors on HuhT7 cells. HuhT7 cells were treated with or without polymerase inhibitors for 48 h. Cell viability was measured by dimethylthiazol carboxymethoxyphenyl sulfophenyl tetrazolium (MTS) assays. (**A**) Favipiravir, (**B**) ribavirin, (**C**) clemizole, (**D**) lomibuvir, (**E**) PSI-6206, (**F**) sofosbuvir, (**G**) foscarnet sodium, (**H**) valacyclovir hydrochloride, (**I**) vidarabine, (**J**) oltipraz, (**K**) zalcitabine, (**L**) clevudine, (**M**) famciclovir, (**N**) tenofovir, (**O**) salicylanilide, (**P**) adefovir dipivoxil, (**Q**) entecavir hydrate, and (**R**) lamivudine. Data are expressed as the means and standard deviations of triplicate determinations from three independent experiments. Statistical significance was determined using a two-tailed Student’s *t* test.

**Figure 2 ijms-23-02631-f002:**
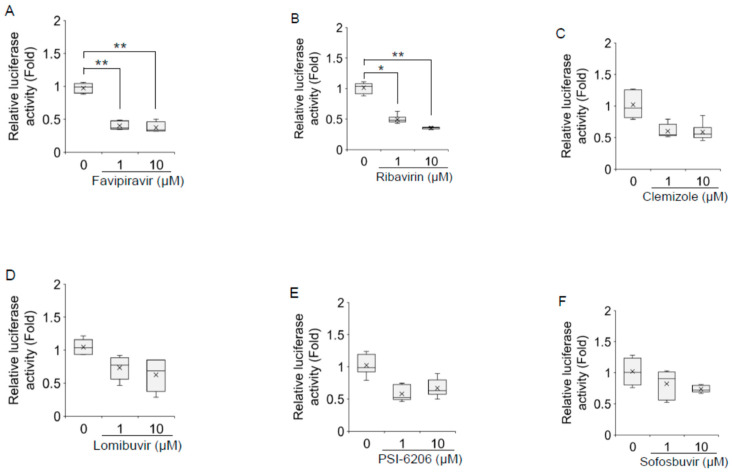
Effects of 18 polymerase inhibitors on hepatitis A virus (HAV) subgenomic replicon replication in HuhT7 cells. HuhT7 cells transfected with the HAV subgenomic replicon were treated with or without polymerase inhibitors for 48 h, and reporter activity was measured by a relative luciferase assay after 72 h of transfection. (**A**) Favipiravir, (**B**) ribavirin, (**C**) clemizole, (**D**) lomibuvir, (**E**) PSI-6206, (**F**) sofosbuvir, (**G**) foscarnet sodium, (**H**) valacyclovir hydrochloride, (**I**) vidarabine, (**J**) oltipraz, (**K**) zalcitabine, (**L**) clevudine, (**M**) famciclovir, (**N**) tenofovir, (**O**) salicylanilide, (**P**) adefovir dipivoxil, (**Q**) entecavir hydrate, and (**R**) lamivudine. Data are presented as the means and standard deviations of triplicate determinations from at least three independent experiments. Statistical significance was determined using a two-tailed Student’s *t* test: * *p* < 0.05, ** *p* < 0.01.

**Figure 3 ijms-23-02631-f003:**
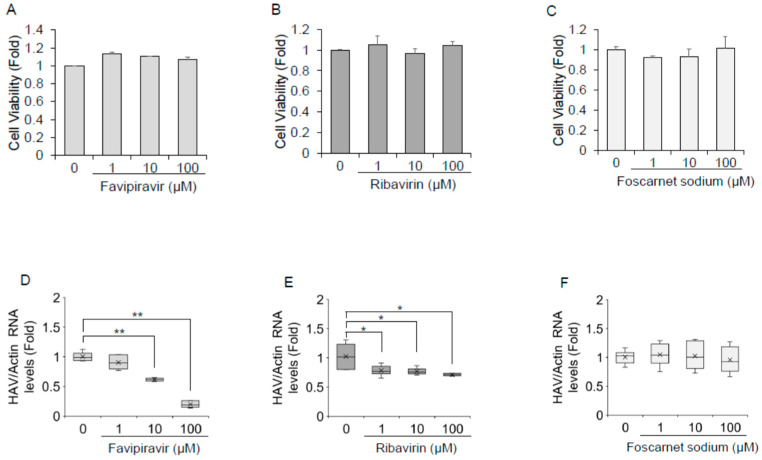
Favipiravir and ribavirin inhibit hepatitis A virus (HAV) infection in Huh7 cells (**A**–**C**). The viability of Huh7 cells was assessed after being treated with favipiravir (**A**), ribavirin (**B**), and foscarnet sodium (**C**) for 72 h. Cell viability was measured by dimethylthiazol carboxymethoxyphenyl sulfophenyl tetrazolium (MTS) assays. Panel D–F. Huh7 cells infected with the HAV HA-11-1299 genotype IIIA strain were treated with favipiravir (**D**), ribavirin (**E**), and foscarnet sodium (**F**) at 0, 1, 10, and 100 μM for 72 h. HAV RNA levels were examined by real-time reverse transcription polymerase chain reaction. Actin mRNA was used as an internal control. HAV RNA levels were downregulated in favipiravir- and ribavirin-treated Huh7 cells. Data are expressed as the means and standard deviations of triplicate determinations from three independent experiments. Statistical significance was determined using a two-tailed Student’s *t* test: * *p* < 0.05, ** *p* < 0.01.

**Figure 4 ijms-23-02631-f004:**
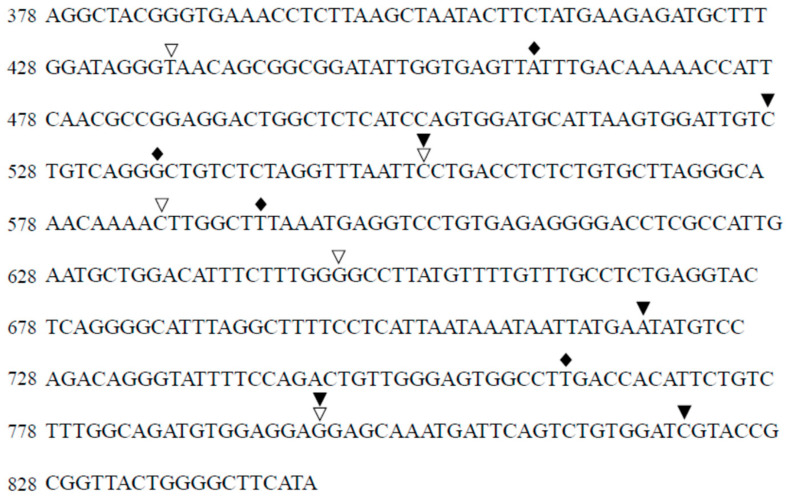
Distribution of more than 80% of mutations that were found in the hepatitis A virus (HAV) 5’ untranslated region (5’ UTR) compared to the untreated control. Introduction of mutagenesis of the HAV 5’ UTR in Huh7 cells treated with or without 100 μM favipiravir (black rhombus), 100 μM ribavirin (black triangle), and 100 μM foscarnet sodium (white triangle). There were no particular mutation hotspots. Almost all mutations were single-nucleotide mutations. Nucleotide number is based on sequence number.

**Figure 5 ijms-23-02631-f005:**
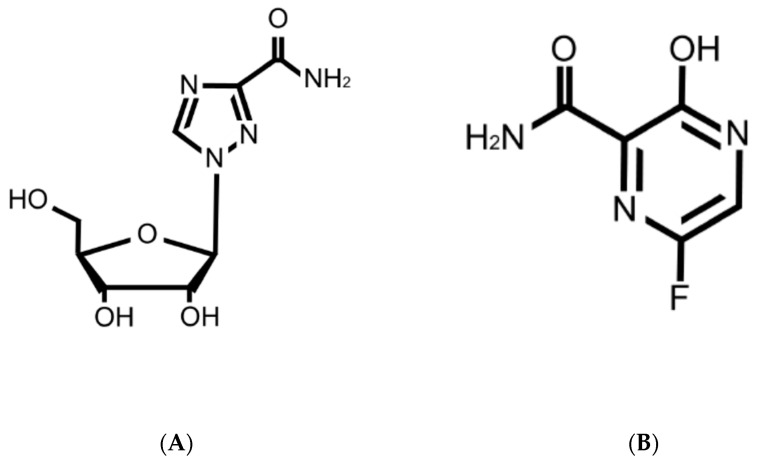
Structures of ribavirin (**A**) and favipiravir (**B**).

**Figure 6 ijms-23-02631-f006:**
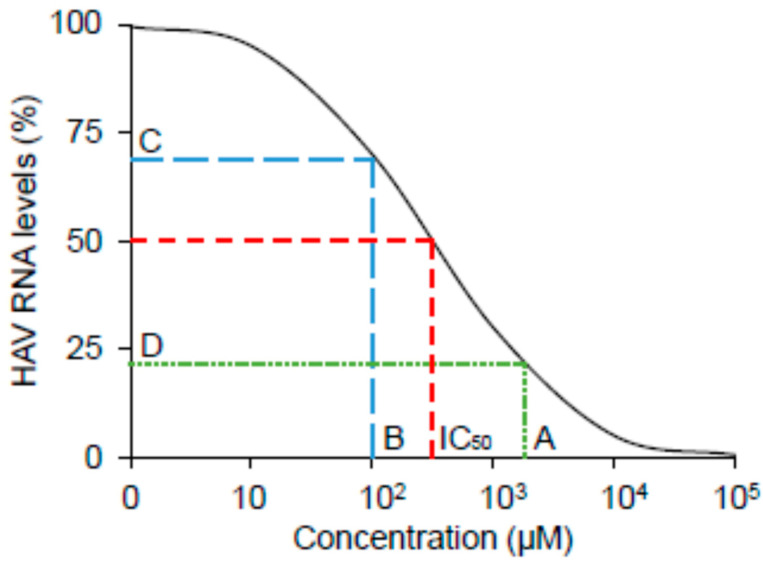
Schematic image of half maximal inhibitory concentration (IC_50_). A higher concentration of two values that sandwich IC_50_ (A), a lower concentration of two values that sandwich IC_50_ (B), HAV RNA levels (%) at B (C), and HAV RNA levels (%) at A (D).

**Table 1 ijms-23-02631-t001:** Number of hepatitis A virus genome sequence mutations derived from Huh7 cells treated with or without favipiravir, ribavirin, and foscarnet sodium.

	Control		Favipiravir		Ribavirin		Foscarnet Sodium	
	N	%	N	%	N	%	N	%
Total	85,107	100	78,253	100	86,047	100	82,399	100
Consensus	65,673	77.2	52,565	67.2	67,054	77.9	62,359	75.7
Mutation	19,434	22.8	25,688	32.8	18,993	22.1	20,040	24.3

N, number of reads; %, % of number of read counts; total (consensus/mutation), total (consensus/mutation) reads.

**Table 2 ijms-23-02631-t002:** PCR primers for real-time reverse transcription polymerase chain reaction (RT-PCR) and deep sequencing in the present study.

Target Gene	Direction	Primer Sequence (5’-3’)
*PCR primers for real-time RT-PCR*		
HAV	Sense	AGGCTACGGGTGAAACCTCTTAG
	Antisense	GCCGCTGTTACCCTATCCAA
Actin	Sense	CAGCCATGTACGTTGCTATCCAGG
	Antisense	AGGTCCAGACGCAGGATGGCATG
*Linker-added specific primers for deep sequencing*		
HAV	Sense	TCGTCGGCAGCGTCAGATGTGTATAAGAGACAGAGGCTACGGGTGAAACCTCTT
	Antisense	GTCTCGTGGGCTCGGAGATGTGTATAAGAGACAGTATGAAGCCCCAGT

Underline indicates the linker portion.

## Data Availability

The data underlying this article are available in this article.

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
