# Peer review of "Favipiravir Inhibits Hepatitis A Virus Infection in Human Hepatocytes"

_ijms, 2022, doi:10.3390/ijms23052631_

Round 1
Reviewer 1 Report
In this manuscript, authors reported that favipiravir inhibits hepatitis A virus replication through the mutational effects. All experiments were carefully performed and well conducted to draw the conclusion. However, there are some problems and lacks as following:
- There are some previous studies about lethal mutagenesis of hepatitis C virus induced by favipiravir (PLOS ONE, October 18, 2016; DOI:10.1371/journal.pone.0164691). Ribavirin also has been reported mutagenic effects on hepatitis C virus. This previous studies are similar with this manuscript except for hepatitis C virus. This point diminish the novelty and significance of the present study. Authors should add more references and describe about that.
- Authors performed advanced experiments for three drugs such as favipiravir, ribavirin, and foscarnet sodium. Why authors wrote just favipiravir in the title.
- In the figure 2, most of drugs had inhibitory effects on HAV subgenomic replication statistically. Authors should check raw data and statistical analysis.
- Authors should conduct further studies of selected drugs for the evaluation of direct antiviral activity (plaque reduction assay etc.)
Author Response
To Reviewer 1
Thank you for your encouraging comment. Accordingly, we also made corrections of grammatical and minor spell errors with red color font.
Response to your comment 1: “There are some previous studies about lethal mutagenesis of hepatitis C virus induced by favipiravir (PLOS ONE, October 18, 2016; DOI:10.1371/journal.pone.0164691). Ribavirin also has been reported mutagenic effects on hepatitis C virus. This previous studies are similar with this manuscript except for hepatitis C virus. This point diminish the novelty and significance of the present study. Authors should add more references and describe about that.”
Thank you for your valuable comments. According to your suggestions, we added new references and revised our manuscript as follows.
In Discussion section, lines 195-212,
HCV and parainfluenza-3 virus were 20.9 μM, 12.8 μM and 197.9 μM, respectively [19, 26]. In the present study, the use of 100 μM ribavirin did not show an IC50 (Figure 3). Serum concentration of ribavirin in complete responders during the combination of interferon alfa-2b and ribavirin therapy for chronic hepatitis C was reported to be 2000 – 3000 ng/mL (8 – 12 μM) [27]. We may need to use a much higher concentration of ribavirin for HAV infection in vitro. As higher serum concentration of ribavirin could lead to adverse events, such as hemolytic anemia, it may be difficult for human to use higher dose of ribavirin. Ribavirin inhibits host inosine monophosphate dehydrogenase (IMPDH), which lowers intracellular GTP pools [28]. Moreover, ribavirin is a mutagenic nucleoside analog in rapidly replicating viruses, inducing error catastrophes and competing with physiological nucleotides for the HCV RdRp active site [29]. Ribavirin also induces lethal mutagenesis and error-prone HCV replication [30–32]. We have reported previously that A → G and G → A mutations increased in comparison with the total number of transition mutations [32]. Ribavirin also induces lethal mutagenesis of poliovirus [33]. In the present study, there was no upregulation of the percentage of mutations or specific single-nucleotide variants during ribavirin treatment in HAV infection (Table 1). The effects of 100 μM ribavirin on HAV RNA levels were not observed in PLC/PRF/5 cells. More incubation time might be required.
In Discussion section, lines 214-236,
Favipiravir has been shown to have a broad spectrum against RNA viruses, such as Influenza virus, Ebola virus (EBOV), Crimean-Congo hemorrhagic fever virus, Lassa virus, Rift Valley fever virus, hemorrhagic fever arenavirus, Chikungunya virus and Norovirus [34]. Moreover, favipiravir was effective in reducing viral replication in a recent outbreak of coronavirus disease 2019 (COVID-19) caused by the novel coronavirus designated severe acute respiratory syndrome coronavirus 2 (SARS-CoV-2), and the half-maximal effective concentration (EC50) was 61.88 μM in vitro [35]. The IC50 values for favipiravir against Zika virus and Crimean-Congo hemorrhagic fever were 35 ±14 μM and 1.1 µg/ml, respectively [36, 37]. Oral favipiravir (300 mg/kg dosed once a day) led to 43.5% survival of Ebola virus infection in guinea pigs [38]. Median serum concentration of favipiravir among COVID-19 patients treated with favipiravir was 35.22 – 60.85 μg/mL (224 –387μM) [39]. In the present study, we showed that the IC50 of favipiravir was 18.8 μM for HAV infection (Figure 3). It might be useful and safety for human to use the present dose of favipiravir. Favipiravir is also a mutagenic nucleoside analog (Figure 5, Panel B), and viral RNA polymerase mistakenly recognizes favipiravir-RTP as a purine nucleotide [40]. Increases in C → U or A → G minority single-nucleotide variants during favipiravir treatment have been reported [41, 42]. Another group reported that favipiravir leads to an excess of G → A and C → U transitions as lethal mutagenesis into HCV populations and works as an antivirals against HCV [43]. In the present study, the percentage of mutation was elevated 10% compared to the control (Table 1); however, there were no specific single-nucleotide variants in the HAV 5' UTR during ribavirin or favipiravir treatment (Figure 4). It is possible that favipiravir may induce lethal mutations in the HAV genome and decrease HAV replication.
Response to your comment 2: “Authors performed advanced experiments for three drugs such as favipiravir, ribavirin, and foscarnet sodium. Why authors wrote just favipiravir in the title.”
Thank you for your valuable comments. Favipiravir inhibited HAV replication in both Huh7 and PLC/PRF/5 cells although ribavirin inhibited HAV replication in only Huh7 cells. Next-generation sequencing demonstrated favipiravir could introduce nucleotide mutations into the HAV genome more than ribavirin. Foscanet sodium significantly downregulated HAV subgenomic replicon replication in a dose-dependent manner, however, 100 μM foscarnet sodium did not inhibit HAV replication in HAV-infected Huh7 cells. According to your suggestion, we made a correction of title as follows. “Favipiravir inhibits hepatitis A virus infection in human hepatocytes”
We also examined whether HAV RNA levels could be inhibited by three drugs in PLC/PRF/5 cells. We observed evident effects of favipiravir not but ribavirin or foscarnet sodium in PLC/PRF/5 cells. We revised our manuscript as follows.
In Result section, lines 132-135,
We also examined whether HAV RNA levels could be inhibited by favipiravir or ribavirin in PLC/PRF/5 cells. Favipiravir at a concentration of 100 μM resulted in 30 – 40% reduction in the HAV RNA levels in PLC/PRF/5 cells, whereas HAV RNA levels was not reduced by 100 μM ribavirin in PLC/PRF/5 cells.
Response to your comment 3: “In the figure 2, most of drugs had inhibitory effects on HAV subgenomic replication statistically. Authors should check raw data and statistical analysis.”
Thank you for your valuable suggestions. According to your suggestions, we revised Figure 2, using box and whisker charts (please see the reviewer 2 comments and response.)
Response to your comment 4: “Authors should conduct further studies of selected drugs for the evaluation of direct antiviral activity (plaque reduction assay etc.)”
Thank you for your valuable comments. According to your suggestions, we tried to perform the plaque assay using HAV HA11-1299 genotype IIIA and African green monkey kidney GL37 cells or immortalized human hepatocytes IHH. However, we did not observe any plaque after 14 days of HAV infection in our condition. We think that one of the causes is that HAV HA11-1299 genotype IIIA is not cytopathic strain, because we observed obvious plaques due to VSV in IHH cells (Kanda T, et al. J Virol. 2007 Nov;81(22):12375-81.) and others reported HAV plaque assay in African green monkey kidney BSC-1 cells (Lemon SM, et al. J Clin Microbiol. 1983 May;17(5):834-9.). We will try to perform plaque reduction assay of selected drugs in another future study.

Reviewer 2 Report
General comments:
- avoid "there is" & "These results" expressions in scientific writing.
- avoid using abbreviations in titles and subtitles.
- Do not start a sentence with an abbreviation.
Abstract:
- define RNA and DNA abbreviations.
- it is not clear what "these drugs" means.
- define ALF abbreviation.
- the conclusions must be supported by your reported results.
Define all abreviations in Keywords.
Introduction:
- State of the art is not appropriately presented.
- "spontaneously recover with or without supportive care" ... "spontaneously recover" imply no intervention.
- "However, in cases of acute liver failure" associated to HAV?
- "Reported risk factors ..." it is not clear how high is the risk associated to the listed factors.
- "As of 2020, it is estimated that there are 728 million persons" how is this link with HAV?
- "the number of persons without HAV immunity" this information is not supported by references ...
- "no specific antiviral therapy for HAV infection worldwide" this is not true; please see https://www.ncbi.nlm.nih.gov/pmc/articles/PMC4663202/ and/or https://www.ncbi.nlm.nih.gov/pmc/articles/PMC8540458/
- information starting with line 46 til 74 belongs to the methods section.
- Present also in this section previous researchers on antiviral therapy for HAV infection. Highlight the gaps that support the need of new studies.
- End this section with the aim of your study.
Results
- Column charts to indicate a point estimator (mean) is not appropriate. Use instead box and whisker charts.
- "We then examined the effects of each drug on HAV subgenomic replicon replication in HuhT7 cells." this is duplicate information; please delete.
- "Cells were transiently transfected with 0.2 μg HAV subgenomic replicon using Effectene Transfection Reagent (Qiagen), and after 24 h of transfection, cells were treated with the 18 polymerase inhibitors at 0, 1 and 10 μM. After 72 h of transfection, luciferase activity was determined." this information belongs to the Materials and Methods section.
- "We next evaluated the cytotoxic effects of favipiravir, ribavirin and foscarnet sodium on Huh7 cells by MTS assays." & "We next measured the expression of HAV RNA by real-time RT-PCR in Huh7 cells." uninformative text; please delete.
- "Cell viabilities after treatment with the 3 drugs were exam-124 ined and compared to DMSO-treated controls." thsi information belongs to the Materials and Methods section.
- "Cells were infected with HAV HA11-1299 genotype IIIA and treated with favipiravir, ribavirin or foscarnet sodium at 0, 1, 10, and 100 μM for 72 h" this information belongs to the Materials and Methods section.
- "Together, these results showed that" redundant information; please delete.
- lines 137-140 is a discussion.
- lines 153-163 belongs to Materials and Methods section.
- "%" symbol must stay in the definition of the column - Table 2.
Discussion:
- It is useful to begin the discussion by briefly summarizing the main findings, and explore possible mechanisms or explanations for these findings.
- Emphasize the new and important aspects of your study and put your findings in the context of the totality of the relevant evidence.
- Cite your own results presented in Figures.
- State the limitations of your study, and explore the implications of your findings for future research and for clinical practice or policy.
Materials and methods:
- CO2 subscript is missing.
- It is not clear how luciferase activity was determined and how the results can be interpreted.
- It is not clear how cell growth and cell viability were measured.
- "evaluation of cell growth" is listed in Methods section but no results are reported; please report also cell growth results.
- Provide a picture to exemplify how IC50 was calculated.
- mean and standard deviations are not appropriate for all distributions.
- Anova or Friedman test are apropriate with post hoc analysis is needed.
Conclusions:
- You mainly have one conclusion so the plural is not appropriate
- lines 325-328 must be moved to the discussion section.
- "in vivo" must stay in italics.
Author Response
To Reviewer 2
We thank the reviewer for helpful comments in improving our manuscript. We have responded to each of the critiques and incorporated changes as necessary in the appropriate contexts of the revised manuscript. A point-by-point discussion of the comments from the reviewer is provided below. Accordingly, we also made corrections of grammatical and minor spell errors with red color font.
General comments:
Response to your comment 1: “avoid "there is" & "These results" expressions in scientific writing.”
Thank you for your valuable comments. We made corrections as the reviewer suggested.
Response to your comment 2: “avoid using abbreviations in titles and subtitles.”
Thank you for your valuable comments. We avoided using abbreviations in titles and subtitles as the reviewer suggested.
Response to your comment 3: “Do not start a sentence with an abbreviation.”
Thank you for your valuable comments. We avoided a sentence with an abbreviation and made corrections as the reviewer suggested.
Abstract:
Response to your comment 4:” define RNA and DNA abbreviations.”
Thank you for your valuable comments. We define RNA and DNA abbreviations as ribonucleic acid and deoxyribonucleic acid, respectively. We added this information lines 14, 36 and 71 in the revised manuscript.
Response to your comment 5: “it is not clear what "these drugs" means.”
Thank you for your valuable comments. Accordingly, we revised our manuscript as follows.
In Abstract section, lines 15-18 of the revised manuscript,
… acid polymerase. In the present study, we examined the anti-HAV activity of 18 drugs by measuring the HAV subgenomic replicon and HAV HA11-1299 genotype IIIA replication in human hepatoma cell lines, using reporter assay and real-time reverse transcription-polymerase chain reaction, respectively. Mutagenesis of…
We have cleared that "these drugs" means 18 ribonucleic acid (RNA)-dependent RNA polymerase and RNA-dependent deoxyribonucleic acid polymerase drugs in the manuscript. We added this information line 15 in the revised manuscript.
Response to your comment 6: “define ALF abbreviation.”
Thank you for your valuable comments. We revised our manuscript as follows.
In Abstract section, lines 11-12 of the revised manuscript,
Hepatitis A virus (HAV) is a causative agent of acute hepatitis, and can occasionally induce acute liver failure. However,…
We also defined acute liver failure as ALF in line 40 in the Introduction section of the revised manuscript.
Response to your comment 7: “the conclusions must be supported by your reported results.”
Thank you for your valuable comments. We revised our manuscript as the reviewer suggested as follows.
In Abstract section, lines 23-26 of the revised manuscript,
…than ribavirin. In conclusion, favipiravir could introduce nucleotide mutations into the HAV genome and work as an antiviral against HAV infection. Provided that further in vivo experiments confirm its efficacy, favipiravir would be useful for the treatment of severe HAV infection.
Response to your comment 8: “Define all abreviations in Keywords”.
Thank you for your valuable comments. We made corrections in Keywords as the reviewer suggested.
Introduction:
Response to your comment 9: “State of the art is not appropriately presented.”
Thank you for your valuable comments. We extensively revised introduction section as the reviewer suggested.
Response to your comment 10: "spontaneously recover with or without supportive care" ... "spontaneously recover" imply no intervention.
Thank you for your valuable comments. Sure. the reviewer is right. We made corrections as follows.
In Introduction section, line 39,
In general, individuals with HAV infection recover with or without intervention. However,…
Response to your comment 11: "However, in cases of acute liver failure" associated to HAV?
Thank you for your valuable comments. Sure. We revised our manuscript as the reviewer suggested.
Response to your comment 12: "Reported risk factors ..." it is not clear how high is the risk associated to the listed factors.
Thank you for your valuable comments. Accordingly, we revised our manuscript as follows.
In Introduction section, lines 44-47,
Adjusted odds ratio for death by aged 40-59 years and aged over 60 years were 7.89 and 14.88, respectively, compared to aged 0-19 years [3]. Adjusted odds ratio for death by preexisting nonviral liver disease, history of hepatitis B, diabetes and cardiovascular disease were 5.2, 2.4, 2.2 and 2.2, respectively [4].
Response to your comment 13: "As of 2020, it is estimated that there are 728 million persons" how is this link with HAV?
Thank you for your valuable comments. Accordingly, we revised our manuscript as follows.
In Introduction section, lines 44-51,
…[3, 4]. Adjusted odds ratio for death by aged 40-59 years and aged over 60 years were 7.89 and 14.88, respectively, compared to aged 0-19 years [3]. Adjusted odds ratio for death by preexisting nonviral liver disease, history of hepatitis B, diabetes and cardiovascular disease were 5.2, 2.4, 2.2 and 2.2, respectively [4].
As of 2020, it is estimated that there are 728 million persons aged 65 years or older worldwide, and this number is expected to increase to 1.5 billion older persons by 2050 [5]. So, it is possible that the number of patients with HAV-associated ALF would increase in near future.
Response to your comment 14: "the number of persons without HAV immunity" this information is not supported by references ...”
Thank you for your valuable comments. We have included new reference [7].
- Yan, J.; Kanda, T.; Wu, S.; Imazeki, F.; Yokosuka, O. Hepatitis A, B, C and E virus markers in Chinese residing in Tokyo, Japan. Hepatol Res. 2012, 42(10), 974–981. doi: 10.1111/j.1872-034X.2012.01009.x. PMID: 22524475
Response to your comment 15: "no specific antiviral therapy for HAV infection worldwide" this is not true; please see https://www.ncbi.nlm.nih.gov/pmc/articles/PMC4663202/ and/or https://www.ncbi.nlm.nih.gov/pmc/articles/PMC8540458/”
Thank you for your valuable suggestion. We have included new references [8] and [9], and we revised our manuscript accordingly.
Response to your comment 15:” information starting with line 46 til 74 belongs to the methods section.”
Thank you for your valuable suggestion. We revised our manuscript as the reviewer suggested.
Response to your comment 16: “Present also in this section previous researchers on antiviral therapy for HAV infection. Highlight the gaps that support the need of new studies.”
Thank you for your valuable comments. Accordingly, we revised our manuscript as follows.
In Introduction section, lines 54-68,
…without HAV immunity will increase with improvements in hygiene [7], thus, there is no doubt that antiviral therapies for HAV infection will be urgently required. As these factors also may make it possible that the number of patients with severe HAV infection could increase, it is important to take appropriate measures promptly against HAV-associated ALF, including specific antiviral therapies for HAV infection.
Although there have been several reports about specific antiviral therapies for HAV infection, both direct-acting antiviral agents (DAAs) and host-targeting agents (HTAs) to control effectively HAV infection should continue to be explored [8, 9]. Small interfering RNAs against HAV and HAV 3C cysteine protease inhibitors are promising DAAs against HAV [8,10]. Interferons, ribavirin and amantadine are also reported as broad-target HTAs against HAV infection [8, 11, 12]. It is unknown whether these drugs have enough effects in clinical settings, and no specific and potent anti-HAV drug is yet available on the market to date. Development of drugs for HAV infection is challenging now since there were an estimated 170 million new cases of acute hepatitis A [13] and HAV is still the most common cause of acute viral hepatitis [14].
In the present study, therefore, …
Response to your comment 17: “End this section with the aim of your study.”
Thank you for your valuable suggestion. Aim of our study is to investigate potentially effective drugs by drug repositioning, and explore the mechanism of action of selected drugs. We made corrections as the reviewer suggested and describe the aim of our study at the end of “Introduction section”.
Results
Response to your comment 18: “Column charts to indicate a point estimator (mean) is not appropriate. Use instead box and whisker charts.”
Thank you for your valuable suggestion. We made a new Figures 2 and 3(D)-(F), using instead box and whisker charts as the reviewer suggested.
Response to your comment 19: “"We then examined the effects of each drug on HAV subgenomic replicon replication in HuhT7 cells." this is duplicate information; please delete.”
Thank you for your valuable suggestion. We deleted this sentence as the reviewer suggested.
Response to your comment 20:”"Cells were transiently transfected with 0.2 μg HAV subgenomic replicon using Effectene Transfection Reagent (Qiagen), and after 24 h of transfection, cells were treated with the 18 polymerase inhibitors at 0, 1 and 10 μM. After 72 h of transfection, luciferase activity was determined." this information belongs to the Materials and Methods section.”
Thank you for your valuable suggestion. We moved this information to the Materials and Methods section as the reviewer suggested.
Response to your comment 21: “"We next evaluated the cytotoxic effects of favipiravir, ribavirin and foscarnet sodium on Huh7 cells by MTS assays." & "We next measured the expression of HAV RNA by real-time RT-PCR in Huh7 cells." uninformative text; please delete.”
Thank you for your valuable suggestion. We deleted this sentence as the reviewer suggested.
Response to your comment 22: “"Cell viabilities after treatment with the 3 drugs were exam-124 ined and compared to DMSO-treated controls." thsi information belongs to the Materials and Methods section.”
Thank you for your valuable suggestion. We moved this information to the Materials and Methods section as the reviewer suggested.
Response to your comment 23: “"Cells were infected with HAV HA11-1299 genotype IIIA and treated with favipiravir, ribavirin or foscarnet sodium at 0, 1, 10, and 100 μM for 72 h" this information belongs to the Materials and Methods section.”
Thank you for your valuable suggestion. We moved this information to the Materials and Methods section as the reviewer suggested.
Response to your comment 24: “"Together, these results showed that" redundant information; please delete.”
Thank you for your valuable suggestion. We deleted this sentence as the reviewer suggested.
Response to your comment 25: ”lines 137-140 is a discussion.”
Thank you for your valuable suggestion. We moved this information to the Discussion section as the reviewer suggested.
Response to your comment 26:” lines 153-163 belongs to Materials and Methods section.”
Thank you for your valuable suggestion. We moved this information to the Materials and Methods section as the reviewer suggested.
Response to your comment 27: ”"%" symbol must stay in the definition of the column - Table 2.”
Thank you for your valuable suggestion. We made corrections in Table 2 legend as the reviewer suggested.
Discussion:
Response to your comment 28: “It is useful to begin the discussion by briefly summarizing the main findings, and explore possible mechanisms or explanations for these findings.”
Thank you for your valuable suggestion. We have added the briefly summarizing the main findings and explanations for these findings in the revised manuscript lines 177-183 as follows.
In the present study, we focused on the effects of 6 RdRp inhibitors and 12 RdDp inhibitors on HAV replication to investigate therapeutic antiviral drugs through drug repositioning. We demonstrated the anti-HAV activity of favipiravir and ribavirin by measuring the HAV subgenomic replicon in HuhT7 and HAV replication in Huh7 cells. We also investigated mutagenesis of the HAV 5' UTR using next-generation sequencing methods. We found that favipiravir decreased HAV replication in a dose-dependent manner and induced nucleotide mutations in the HAV genome (Table 1). As a catalytic domain of RdRp is widely…
Response to your comment 29: “Emphasize the new and important aspects of your study and put your findings in the context of the totality of the relevant evidence.”
Thank you for your valuable suggestion. We would like to emphasize that favipiravir could introduce nucleotide mutations into the HAV genome and work as an antiviral against HAV infection. We extensively revised abstract and discussion section accordingly.
Response to your comment 30: “Cite your own results presented in Figures.”
Thank you for your valuable suggestion. Sure. We revised our manuscript accordingly.
Response to your comment 31: “State the limitations of your study, and explore the implications of your findings for future research and for clinical practice or policy.”
Thank you for your valuable suggestion. Sure. We made corrections as follows.
In Discussion section, lines 259-266,
…On this point, further investigation is needed.
In the present study, Huh7 and its derived cells were used for most of experiments. As this may be one of the study’s limitations, another cell lines would be useful for further study [50]. As another limitation of the present study, we did not examine the direct effects of favipiravir on HAV infection using plaque assay [51, 52]. Further studies focusing on the mutagenesis of HAV polymerase genes and the meaning of these mutations, and in vivo evaluation of favipiravir administration in animal models, will also be important to assess the therapeutic effects of favipiravir. After these evidences were established, we examined whether favipiravir is effective for HAV infection with safety.
Materials and methods:
Response to your comment 32: “CO2 subscript is missing.”
Thank you for your valuable suggestion. We made corrections as the reviewer suggested.
Response to your comment 33:”It is not clear how luciferase activity was determined and how the results can be interpreted.”
Thank you for your valuable suggestion. According to your suggestion, we revised our manuscript as follows.
In Materials and Methods section, lines 320-321,
…JNR II AB-2300 (ATTO, Tokyo, Japan). Firefly luciferase activities were compared to DMSO-treated controls.
Response to your comment 34: “It is not clear how cell growth and cell viability were measured.”
Thank you for your valuable suggestion. We determined the cell viability by compering to DMSO-treated controls. We made the corrections as the reviewer suggested in the “Materials and methods” section.
Response to your comment 35: “"evaluation of cell growth" is listed in Methods section but no results are reported; please report also cell growth results.”
Thank you for your valuable suggestion. We deleted "evaluation of cell growth" from “Materials and methods” section.
Response to your comment 36: “Provide a picture to exemplify how IC50 was calculated.”
Thank you for your valuable suggestion. We made a figure 6 as the reviewer suggested, and revised our manuscript.
Response to your comment 37:”mean and standard deviations are not appropriate for all distributions.”
Thank you for your suggestion. We revised Figure 2 and Figure 3 using box and whisker charts as the reviewer suggested.
Response to your comment 38:”Anova or Friedman test are apropriate with post hoc analysis is needed.”
No thanks. After we consulted experts, we think these analyses are not proper in our study.
Conclusions:
Response to your comment 39: “You mainly have one conclusion so the plural is not appropriate”
Thank you for your suggestion. We extensively revised the conclusion section of our manuscript as the reviewer suggested.
Response to your comment 40:” lines 325-328 must be moved to the discussion section.
Thank you for your suggestion. We moved these lines 325-328 to the discussion section.”
Response to your comment 41: "in vivo" must stay in italics.
Thank you for your suggestion. We made the corrections as the reviewer suggested.

Reviewer 3 Report
The authors describe in this manuscript the in vitro evaluation against hepatitis A virus of 18 known RNA- or DNA-polymerase inhibitors. The potential anti-HAV efficacy of these drugs was evaluated by measuring the HAV subgenomic replicon and HAV HA11-1299 genotype IIIA replication and mutagenesis of the HAV 5' untranslated region. The authors can be congratulated for the quality of their work, which is well exposed; the organization of the article is clear and the subject fits quite well to the scope of the journal. Suitable and sufficient bibliographical references illustrate this work properly.
I have only a few remarks and suggestions to bring to the attention of the authors:
- Line 11, rewording suggestion: “Hepatitis A virus (HAV) is a causative agent of acute hepatitis.”
- Line 13, rewording suggestion: “However, no specific and potent anti-HAV drug is available on the market to date. Thus, we investigated several novel therapeutic drugs through a drug repositioning approach, targeting…”
- Line 17, rewording suggestion: “… translated region. These specific parameters were explored because lethal mutagenesis…”
- Line 21, rewording suggestion: “Provided that further in vivo experiments confirm its efficacy, favipiravir would be useful…”
- Lines 43-44, rewording suggestion: “in hygiene, thus, there is no doubt that specific antiviral therapy for HAV infection will be urgently required.”
- Line 45, rewording suggestion: “To this day, there is no specific […] worldwide. Therefore, we investigated in this study potentially effective…”
- Lines 57 and 69: is there a rational to explain the evaluation of valacyclovir and adefovir dipivoxil prodrugs instead of the directly active compounds acyclovir and adefovir? How can we be sure that the HuhT7 cells will properly hydrolyze these ester prodrugs?
- Lines 120 and 151, typo: please write “**P < 0.01.” with spaces before and after the < symbol, to be consistent with “*P < 0.05”
- Lines 210 and 222, typo: please write “IC50” with 50 in subscript.
- Lines 207 and 219-220: specifying the dosages usually used in humans in the management of these diseases could be informative.
- Lines 206 and 226, suggestion left to the discretion of the authors: as it is described at lines 206 and 226, an additional figure showing the chemical structures of ribavirin and favipiravir could be appropriate, positioned before the last paragraph of the Discussion section and with references at line 206 (“… nucleoside analog (Figure 5)”) and 226 (“… nucleoside analog (Figure 5),”
Author Response
To Reviewer 3
Response to your comment 1: “The authors describe in this manuscript the in vitro evaluation against hepatitis A virus of 18 known RNA- or DNA-polymerase inhibitors. The potential anti-HAV efficacy of these drugs was evaluated by measuring the HAV subgenomic replicon and HAV HA11-1299 genotype IIIA replication and mutagenesis of the HAV 5' untranslated region. The authors can be congratulated for the quality of their work, which is well exposed; the organization of the article is clear and the subject fits quite well to the scope of the journal. Suitable and sufficient bibliographical references illustrate this work properly.
We thank the reviewer for the encouraging comments. A point-by-point discussion of the comments from the reviewer is provided below. Accordingly, we also made corrections of grammatical and minor spell errors with red color font.
I have only a few remarks and suggestions to bring to the attention of the authors:
Response to your comment 2: - Line 11, rewording suggestion: “Hepatitis A virus (HAV) is a causative agent of acute hepatitis.”
Thank you for your valuable comments. We revised our manuscript as following. Line 11, “Hepatitis A virus (HAV) is a causative agent of acute hepatitis,”
Response to your comment 3: - Line 13, rewording suggestion: “However, no specific and potent anti-HAV drug is available on the market to date. Thus, we investigated several novel therapeutic drugs through a drug repositioning approach, targeting…”
Thank you for your valuable comments. We made corrections as following. Line 12,
“However, no specific potent anti-HAV drug is available on the market to date. Thus, we investigated several novel therapeutic drugs through a drug repositioning approach, targeting…”
Response to your comment 4: - Line 17, rewording suggestion: “… translated region. These specific parameters were explored because lethal mutagenesis…”
Thank you for your valuable comments. Accordingly, we revised our manuscript as follows.
In Abstract section, lines 18-20,
…, respectively. Mutagenesis of the HAV 5' untranslated region were also examined by next-generation sequencing. These specific parameters were explored because lethal mutagenesis has emerged as a novel potential therapeutic approach to treat RNA virus infections. Favipiravir …
Response to your comment 5: - Line 21, rewording suggestion: “Provided that further in vivo experiments confirm its efficacy, favipiravir would be useful…”
Thank you for your valuable comments. We made corrections as following. Lines 25-26,
“Provided that further in vivo experiments confirm its efficacy, favipiravir would be useful for the treatment of severe HAV infection.”
Response to your comment 6: - Lines 43-44, rewording suggestion: “in hygiene, thus, there is no doubt that specific antiviral therapy for HAV infection will be urgently required.”
Thank you for your valuable comments. We made corrections as following. Line 54,
“in hygiene [7], thus, there is no doubt that antiviral therapies for HAV infection will be urgently required.”
Response to your comment 7: - Line 45, rewording suggestion: “To this day, there is no specific […] worldwide. Therefore, we investigated in this study potentially effective…”
Thank you for your valuable comments. Accordingly, we revised our manuscript as follows.
In Introduction section, lines 64-76,
…against HAV infection [8, 11, 12]. It is unknown whether these drugs have enough effects in clinical settings, and no specific and potent anti-HAV drug is yet available on the market to date. Development of drugs for HAV infection is challenging now since there were an estimated 170 million new cases of acute hepatitis A [13] and HAV is still the most common cause of acute viral hepatitis [14].
In the present study, therefore, we investigated potentially effective drugs by drug repositioning. We examined the anti-HAV activity of these 18 drugs, including 6 RdRp inhibitors and 12 RNA-dependent deoxyribonucleic acid (DNA) polymerase (RdDp) inhibitors, by measuring the HAV subgenomic replicon and HAV replication. To explore the mechanism of action of selected drugs, we also examined the mutagenesis of the HAV 5' UTR, using next-generation sequencing methods. Updated drugs for acute hepatitis A is needed since improved cure rates of acute HAV infection is critical to create strategies for global intervention.
Response to your comment 8: - Lines 57 and 69: is there a rational to explain the evaluation of valacyclovir and adefovir dipivoxil prodrugs instead of the directly active compounds acyclovir and adefovir? How can we be sure that the HuhT7 cells will properly hydrolyze these ester prodrugs?
Thank you for your valuable comments. Accordingly, we revised our manuscript as follows.
In Result section, lines 106-108,
…(Figure 2, Panels C–F and I–R). As we used valacyclovir hydrochloride and adefovir dipivoxil prodrugs instead of the directly active compounds acyclovir and adefovir, it is possible that these ester prodrugs could not be properly hydrolyzed in HuhT7 cells. Our…
Response to your comment 9: - Lines 120 and 151, typo: please write “**P < 0.01.” with spaces before and after the < symbol, to be consistent with “*P < 0.05”
Thank you for your valuable comments. We made corrections as the reviewer suggested.
Response to your comment 10: - Lines 210 and 222, typo: please write “IC50” with 50 in subscript.
Thank you for your valuable comments. We made corrections as the reviewer suggested.
Response to your comment 11: - Lines 207 and 219-220: specifying the dosages usually used in humans in the management of these diseases could be informative.
Thank you for your valuable comments. We added the dosages usually used in humans in the management of favipiravir and ribavirin in Discussion section as following. Line 196,
Thank you for your valuable comments. Accordingly, we revised our manuscript as follows.
In Discussion section, lines 196-200,
…26]. In the present study, the use of 100 μM ribavirin did not show an IC50 (Figure 3). Serum concentration of ribavirin in complete responders during the combination of interferon alfa-2b and ribavirin therapy for chronic hepatitis C was reported to be 2000 – 3000 ng/mL (8 – 12 μM) [27]. We may need to use a much higher concentration of ribavirin for HAV infection in vitro….
In Discussion section, lines 222-227,
…µg/ml, respectively [36, 37]. Oral favipiravir (300 mg/kg dosed once a day) led to 43.5% survival of Ebola virus infection in guinea pigs [38]. Median serum concentration of favipiravir among COVID-19 patients treated with favipiravir was 35.22 – 60.85 μg/mL (224 –387μM) [39]. In the present study, we showed that the IC50 of favipiravir was 18.8 μM for HAV infection (Figure 3). It might be useful and safety for human to use the present dose of favipiravir….
Response to your comment 12: - Lines 206 and 226, suggestion left to the discretion of the authors: as it is described at lines 206 and 226, an additional figure showing the chemical structures of ribavirin and favipiravir could be appropriate, positioned before the last paragraph of the Discussion section and with references at line 206 (“… nucleoside analog (Figure 5)”) and 226 (“… nucleoside analog (Figure 5),”
Thank you for your valuable comments. We made a figure 5 in the revised manuscript as the reviewer suggested.

Round 2
Reviewer 1 Report
I read carefully this revised manuscript entitled “Favipiravir inhibits hepatitis A virus infection in human hepatocytes”. Authors responded and corrected the manuscript for most of the comments I had pointed out. Thus, I recommend this work to be published in the “Int. J. Mol. Sci.”
Author Response
To Reviewer 1
Thank you very much for your encouraging comment.

Reviewer 2 Report
The authors appropriately addressed the comments and suggestions.
It would be great if in the published manuscript Figures 1-3 would be larger.
Author Response
To Reviewer 2
Thank you very much for your encouraging comment.
Response to your comment: “It would be great if in the published manuscript Figures 1-3 would be larger.”
Thank you very much for your valuable suggestions. We revised Figures 1-3.
